Prevalence of poor sleep quality and its associated factors in patients with concurrent type 2 diabetes mellitus and hypertension

Feng Ziling 1
Liu Hongying 2
Xiong Ni 1
Tang Leyao 1
Dai Wenjie 1
Yang Fang 973668791@qq.com 3
1 Department of Epidemiology and Health Statistics, Xiangya School of Public Health, Central South University , Changsha , Hunan Province , China
2 Medical Department, Hunan Prevention and Treatment Institute for Occupational Diseases, Affiliated Prevention and Treatment Institute for Occupational Diseases of University of South China , Changsha , Hunan Province , China
3 Medical Simulation Center, 921 Hospital , Changsha , Hunan Province , China
Lu Frank
Electronic publication date: 2025 Nov 13
Publication date: 2025
Volume: 13
Electronic Location ID: e20325
Received 2025 May 16; Accepted 2025 Oct 13
Copyright: ©2025 Feng et al.
Copyright year: 2025
Copyright holder: Feng et al.
License: This is an open access article distributed under the terms of the Creative Commons Attribution License, which permits unrestricted use, distribution, reproduction and adaptation in any medium and for any purpose provided that it is properly attributed. For attribution, the original author(s), title, publication source (PeerJ) and either DOI or URL of the article must be cited.
License URL: https://creativecommons.org/licenses/by/4.0/

Keywords: Sleep quality, Type 2 diabetes mellitus, Hypertension, Prevalence, Associated factors

Funding: National Natural Science Foundation of China 82103939 National Natural Science Foundation of Hunan Province 2025JJ90224 This study was supported by the National Natural Science Foundation of China under (Grant No. 82103939), and the National Natural Science Foundation of Hunan Province under (Grant No. 2025JJ90224). The funders had no role in study design, data collection and analysis, decision to publish, or preparation of the manuscript.

==============================
Background

The coexistence of type 2 diabetes mellitus (T2DM) and hypertension can impair sleep quality, potentially leading to a wide range of adverse health outcomes. Therefore, this study aimed to evaluate sleep quality and its associated factors in patients with concurrent T2DM and hypertension in Hunan, China.

Methods

This cross-sectional study recruited patients with concurrent T2DM and hypertension who visited the Department of Endocrinology at the First People’s Hospital of Ruanjiang City, Hunan Province, China, between July 2022 and February 2023. Trained investigators conducted face-to-face interviews to collect data on sociodemographic, lifestyle, and disease-related factors, and assessed anxiety and sleep quality using the Generalized Anxiety Disorder-7 scale and the Pittsburgh Sleep Quality Index, respectively. Multivariable logistic regression analysis was performed to identify factors independently associated with sleep quality. A receiver operating characteristic (ROC) curve was generated to evaluate the predictive ability of the model, and the Hosmer–Lemeshow test was used to assess model calibration. Subgroup analyses were conducted by sex, educational level, and number of T2DM complications to test model robustness.

Results

A total of 475 patients with concurrent T2DM and hypertension were included. The prevalence of poor sleep quality was 59.4% (95% confidence interval (CI) [54.9%–63.9%]). Multivariable logistic regression analysis showed that advanced age (adjusted odds ratio (aOR) = 2.12, 95% CI [1.29–3.48]), history of stroke (aOR = 2.16, 95% CI [1.15–4.06]), and anxiety (aOR = 4.24, 95% CI [2.58–6.98]) were associated with higher odds of poor sleep quality. Regular physical activity was associated with lower odds (aOR = 0.53, 95% CI [0.34–0.84]). The area under the receiver operating characteristic (ROC) curve was 0.776 (95% CI [0.735–0.818]), and the Hosmer–Lemeshow test (P = 0.260) indicated good model calibration. Subgroup analyses yielded similar results.

Conclusions

Nearly three-fifths of patients with concurrent T2DM and hypertension exhibited poor sleep quality. Age, regular physical activity, history of stroke, and anxiety were significantly associated with sleep quality. However, due to the cross-sectional design, causal relationships cannot be established.

Introduction

Hypertension and diabetes are two of the 10 most common chronic diseases that significantly contribute to the global economic burden (Mills, Stefanescu & He, 2020; Bommer et al., 2018). In 2021, the global prevalence of diabetes in individuals aged 20–79 years was 10.5%, and it was projected to increase to 12.2% by 2045 (Sun et al., 2022). Furthermore, the number of individuals living with hypertension doubled between 1990 and 2019, from 650 million to 1.3 billion (World Health Organization, 2023). Diabetes and hypertension often co-occur (Ferrannini & Cushman, 2012), and a meta-analysis showed that the pooled prevalence of hypertension in Chinese patients with type 2 diabetes mellitus (T2DM) was 54% (95% confidence interval 95% CI [47%–61%]) (Zhang et al., 2024). Accumulating evidence has shown that the coexistence of diabetes and hypertension can not only increase the risk of cardiovascular events and mortality but also impair sleep quality (Przezak, Bielka & Pawlik, 2022; Petrie, Guzik & Touyz, 2018; Surani et al., 2015; Chiang et al., 2018).

Sleep quality serves as an indicator of how energized, active, and prepared an individual feels for a new day. It can be assessed both subjectively and objectively through quantitative sleep aspects such as sleep latency and sleep duration (Tang et al., 2017; Liu et al., 1996). Due to nocturia, polyuria, neuropathy, and restless legs syndrome, patients with T2DM experience increased nocturnal awakenings and reduced sleep duration (Surani et al., 2015; Barone & Menna-Barreto, 2011; Kuo et al., 2021). Additionally, patients with hypertension may experience nocturnal urination caused by diuretics (Rahman et al., 2021), and uncontrolled hypertension or some antihypertensive drugs may disrupt the parasympathetic regulation and pressure reflex of the heart, which can trigger arousal (Liu et al., 2016; Oliveira-Silva et al., 2020). The concurrence of diabetes and hypertension may have synergistic effects on sleep quality (Jemere et al., 2019; Phan et al., 2023). For example, a study of T2DM patients in Ethiopia found that the concurrence of hypertension can increase the odds of poor sleep quality (adjusted odds ratio [aOR] = 3.19, 95% CI [1.16–8.84]) (Jemere et al., 2019). Similarly, a study by Phan et al. (2023) found that individuals with chronic comorbidities such as diabetes and hypertension had higher odds of poor sleep quality (OR = 2.34, 95% CI [1.17–5.24]). Evidence has consistently shown that poor sleep quality can lead to a wide range of negative health-related outcomes (Miller & Howarth, 2023; Kwok et al., 2018; Krittanawong et al., 2019; Donga et al., 2010; Grandner, 2017). Therefore, understanding the prevalence of poor sleep quality and its associated factors in patients with concurrent T2DM and hypertension is essential for not only early identification of individuals at high risk for poor sleep quality but also finding intervention targets to improve sleep quality.

Previous studies have examined the prevalence of poor sleep quality and its associated factors in patients with diabetes or hypertension alone (Nasir et al., 2022; Birhanu, Hassen Salih & Abate, 2020; Ayanaw et al., 2022; Birhanu et al., 2021). However, significant disparities have been observed across these studies (Nasir et al., 2022; Birhanu, Hassen Salih & Abate, 2020; Li et al., 2020). For example, a study conducted in Malaysia found that the prevalence of poor sleep quality in T2DM patients was 32% (Nasir et al., 2022), whereas a study conducted in Ethiopia found the prevalence was 47.2% (Birhanu, Hassen Salih & Abate, 2020). Moreover, a meta-analysis by Li et al. (2020) showed that the pooled prevalence of poor sleep quality in Chinese hypertension patients was 52.5%. The differences in study regions, study populations, and tools used to assess sleep quality may contribute to the differences in the prevalence of poor sleep quality observed across studies. Socio-demographic and lifestyle factors such as age, sex, smoking and alcohol consumption, as well as disease-related factors such as diabetic comorbidities and anxiety, have been shown to be associated with sleep quality in patients with diabetes or hypertension alone (Birhanu, Hassen Salih & Abate, 2020; Ayanaw et al., 2022; Maimaitituerxun et al., 2024). However, there is still a lack of studies specifically focusing on patients with concurrent T2DM and hypertension in China up to now. Therefore, this study aimed to identify the prevalence of poor sleep quality and its associated factors in patients with concurrent T2DM and hypertension in Hunan, China.

Materials & Methods

Study design and population

This cross-sectional study was conducted from July 2022 to February 2023. Patients with concurrent T2DM and hypertension who visited the Department of Endocrinology at the First People’s Hospital of Ruanjiang City, Hunan Province, China, during the study period were consecutively recruited. The inclusion criteria were (1) diagnosis of T2DM (Society, 2021); (2) diagnosis of hypertension (Chinese Society of Cardiology et al., 2018); (3) aged ≥50; and (4) voluntary participation in this study with a signed informed consent form. Patients with dementia were excluded from this study. The study protocol was approved by the Ethics Committee of Xiangya School of Public Health, Central South University (No: XYGW–2021–27). According to the sample size formula for categorical outcomes in cross-sectional studies “N=Z2p(1-p)/d2” (Bolarinwa, 2020), a minimum sample size of 133 was determined based on the following assumptions: Z1−α/2 = 1.96, p = 42.0% (Dogah et al., 2025), and d = 0.20p.

Data collection

Well-qualified investigators conducted face-to-face interviews to collect data on socio-demographic, lifestyle and disease-related factors, and to assess anxiety status and sleep quality using well-validated scales. Data on disease-related factors were extracted from the electronic medical records. Any missing values were filled immediately following the initial interviews by the investigators.

Study variables

Socio-demographic and lifestyle factors

Socio-demographic and lifestyle factors included age, sex, marital status, educational level, per capita monthly household income, working status, body mass index (BMI), a history of smoking, a history of drinking, and regular physical activity. Specifically, individuals were categorized as having a history of smoking if they had smoked cigarettes cumulatively for at least six months or daily for one cigarette; individuals were categorized as having a history of drinking if they had consumed alcohol at least once a month for over one year; and regular physical activity was defined as performing one or more physical activities (such as walking, cycling or square dancing) for more than 30 min per day in the past month.

Disease-related factors

Disease-related factors included duration of T2DM, duration of hypertension, a family history of T2DM, a family history of hypertension, number of T2DM complications, coronary heart disease, chronic kidney disease, and stroke. Specifically, T2DM complications included diabetic nephropathy, diabetic retinopathy, diabetic foot, diabetic peripheral neuropathy, diabetic peripheral vasculopathy and diabetic ketosis.

Anxiety

The Generalized Anxiety Disorder-7 (GAD-7) by Spitzer et al. (2006) was used to assess anxiety. It involves 7 items scored on a 0–3 scale, and the total score of the GAD-7 scale ranges from 0 to 21. A total score of 4 was used as the cutoff value for identifying anxiety. Specifically, participants with a total score of >4 were considered as positive for anxiety, whereas a total score of ≤4 was considered as not. The GAD-7 was reliable and valid in the Chinese population (Xu et al., 2022).

Study outcome

The outcome of this study was sleep quality, which was assessed using the Pittsburgh Sleep Quality Index (PSQI) scale, developed by Buysse et al. (1989). It involves 18 self-assessment items and 7 dimensions, including subjective sleep quality, sleep latency, sleep duration, habitual sleep efficiency, sleep disturbances, sleep medications, and daytime dysfunction (Buysse et al., 1989). Each dimension is scored on a 0–3 scale, and the total score of the PSQI scale ranges from 0 to 21, with higher scores indicating poorer sleep quality. A total score of 7 was used as the cutoff value for identifying poor sleep quality. Specifically, a total score of >7 was considered as poor sleep quality, whereas a total score of ≤7 was considered as good sleep quality. The PSQI was reliable and valid in the Chinese population (Liu et al., 1996).

Statistical analyses

Means and standard deviations (SD) were used to describe continuous variables that conformed to a normal distribution. Otherwise, medians and interquartile range (IQR) were used. Categorical data were described by the frequencies (n) and proportions (%), and analyzed using the χ2 test or Fisher’s exact probability test as appropriate. Multivariable logistic regression analysis was used to identify the factors independently associated with sleep quality, and all factors with a P value of <0.05 in between-group comparisons were included in the multivariable model. Multicollinearity was assessed using the variance inflation factor (VIF), with a VIF value of <5 considered as the absence of multicollinearity. The receiver operating characteristic (ROC) curve was plotted to test the predictive ability of the established multivariable model using the ROC Analysis Module in SPSS version 26.0 software (IBM Corp., Armonk, NY, USA). The Hosmer-Lemeshow test was used to assess the model calibration, with a P value of >0.05 indicating that the predicted probability of the model was in good agreement with the actual observed values. To assess the robustness of the multivariable model, subgroup analyses were performed according to sex, educational level and number of T2DM complications. All data were analyzed using SPSS version 26.0 software, with a P value of <0.05 indicating statistically significant by two-sided tests.

Results

Characteristics of the study participants

A total of 475 eligible patients with concurrent T2DM and hypertension were included in this study, and no missing data were observed. The sample size of 475 exceeded the minimum requirement for this study (N = 133). Table 1 shows the socio-demographic and lifestyle characteristics of the study participants. The age range was 50–95 years with a mean age of 69.80 ± 9.10 years; 198 (41.7%) were males and 277 (58.3%) were females; 246 (51.8%) attended elementary school at most; 171 (36.0%) had a history of smoking; and 149 (31.4%) had a history of drinking.

Table 1 Socio-demographic and lifestyle factors of participants (n = 475).

Variables	Frequency (n)	Proportion (%)	
Age (years)			
50–69	222	46.7	
≥70	253	53.3	
Sex			
Male	198	41.7	
Female	277	58.3	
Marital status			
Unmarried	64	13.5	
Married	411	86.5	
Educational level			
Elementary school or below	246	51.8	
Middle school	147	30.9	
High school or above	82	17.3	
Per capita monthly household income (yuan)			
≤3,000	302	63.6	
>3,000	173	36.4	
Working status			
No working	430	90.5	
Working	45	9.5	
BMI			
<18.5	36	7.6	
18–24	240	50.5	
24–28	156	32.8	
≥28	43	9.1	
History of smoking			
No	304	64.0	
Yes	171	36.0	
History of drinking			
No	326	68.6	
Yes	149	31.4	
Regular physical activity			
No	256	53.9	
Yes	219	46.1	
Notes.

BMI, body mass index.

Table 2 shows the disease-related characteristics of the study participants. The mean duration of T2DM and hypertension was 13.36 ± 8.42 years and 13.64 ± 8.63 years, respectively; 355 (74.7%) had one or more T2DM complications; 81 (17.1%) had a history of stroke; and 178 (37.5%) were considered to have positive anxiety symptoms.

Table 2 Disease-related factors of participants (n = 475).

Variables	Frequency (n)	Proportion (%)	
Duration of T2DM (years)			
<10	168	35.4	
≥10	307	64.6	
Duration of hypertension (years)			
<10	160	33.7	
≥10	315	66.3	
Family history of T2DM			
No	317	66.7	
Yes	158	33.3	
Family history of hypertension			
No	315	66.3	
Yes	160	33.7	
Number of T2DM complications			
0	120	25.3	
≥1	355	74.7	
Coronary heart disease			
No	254	53.5	
Yes	221	46.5	
Chronic kidney disease			
No	427	89.9	
Yes	48	10.1	
Stroke			
No	394	82.9	
Yes	81	17.1	
Anxiety			
No	297	62.5	
Yes	178	37.5	
Notes.

T2DM, type 2 diabetes mellitus.

Prevalence of poor sleep quality

The mean score of PSQI of the study participants was 9.13 ±  4.81. According to the established cutoff values, 282 and 193 participants were considered as poor and good sleep quality, respectively. The prevalence of poor sleep quality in patients with concurrent T2DM and hypertension was 59.4% (95% CI [54.9%–63.9%]), and this estimate was 64.3% (95% CI [60.0%–68.6%]) and 52.5% (95% CI [40.8%–57.0%]) in female and male participants, respectively.

Univariable analyses of factors associated with sleep quality

The results of the univariable analyses are shown in Tables 3 and 4. Age (P <  0.001), sex (P = 0.010), marital status (P = 0.029), educational level (P < 0.001), per capita monthly household income (P < 0.001), work status (P = 0.005), history of smoking (P = 0.041), regular physical activity (P <  0.001), duration of T2DM (P = 0.036), coronary heart disease (P = 0.006), stroke (P < 0.001), and anxiety (P < 0.001) differed significantly between the poor and good sleep quality groups.

Table 3 Univariable associations of socio-demographic and lifestyle factors with sleep quality.

Variables	Sleep quality	χ 2	P value	Crude OR (95% CI)	
	Good (n = 193, %)	Poor (n = 282, %)				
Age (years)						
50–69	121 (62.7)	101 (35.8)	33.25	<0.001	1	
≥70	72 (37.3)	181 (64.2)			3.01 (2.06–4.40)	
Sex						
Male	94 (48.7)	104 (36.9)	6.59	0.010	1	
Female	99 (51.3)	178 (63.1)			1.63 (1.12–2.36)	
Marital status						
Unmarried	18 (9.3)	46 (16.3)	4.80	0.029	1	
Married	175 (90.7)	236 (83.7)			0.53 (0.30–0.94)	
Educational level						
Elementary school or below	77 (39.9)	169 (59.9)	19.36	<0.001	1	
Middle school	71 (36.8)	76 (27.0)			0.49 (0.32–0.74)	
High school or above	45 (23.3)	37 (13.1)			0.38 (0.23–0.63)	
Per capita monthly household income (Yuan)						
≤3,000	104 (53.9)	198 (70.2)	13.19	<0.001	1	
>3,000	89 (46.1)	84 (29.8)			0.50 (0.34–0.73)	
Working status						
No working	166 (86.0)	264 (93.6)	7.73	0.005	1	
Working	27 (14.0)	18 (6.4)			0.42 (0.22–0.79)	
BMI						
<18.5	12 (6.2)	24 (8.5)	1.24	0.743	1	
18.5–24	96 (49.8)	144 (51.1)			0.75 (0.36–1.57)	
24–28	66 (34.2)	90 (31.9)			0.68 (0.32–1.46)	
≥28	19 (9.8)	24 (8.5)			0.63 (0.25–1.58)	
History of smoking						
No	113 (58.5)	191 (67.7)	4.19	0.041	1	
Yes	80 (41.5)	91 (32.3)			0.67 (0.46–0.98)	
History of drinking						
No	124 (64.2)	202 (71.6)	2.90	0.089	1	
Yes	69 (35.8)	80 (28.3)			0.71 (0.48–1.05)	
Regular physical activity						
No	67 (34.7)	189 (67.0)	48.13	<0.001	1	
Yes	126 (65.3)	93 (33.0)			0.26 (0.18–0.39)	
Notes.

OR odds ratio

95% CI 95% confidence interval

BMI body mass index

Table 4 Univariable associations of disease-related factors with sleep quality.

Variables	Sleep quality	χ 2	P value	Crude OR (95% CI)	
	Good (n = 193, %)	Poor (n = 282, %)				
Duration of T2DM (years)						
<10	79 (40.9)	89 (31.6)	4.40	0.036	1	
≥10	114 (59.1)	193 (68.4)			1.50 (1.03–2.20)	
Duration of hypertension (years)						
<10	74 (38.3)	86 (30.5)	3.16	0.076	1	
≥10	119 (61.7)	196 (69.5)			1.42 (0.96–2.08)	
Family history of T2DM						
No	127 (65.8)	190 (67.4)	0.13	0.721	1	
Yes	66 (34.2)	92 (32.6)			0.93 (0.63–1.37)	
Family history of hypertension						
No	131 (67.9)	184 (65.2)	0.35	0.552	1	
Yes	62 (32.1)	98 (34.8)			1.13 (0.76–1.66)	
Number of T2DM complications						
0	43 (22.3)	77 (27.3)	1.52	0.216	1	
≥1	150 (77.7)	205 (72.7)			0.76 (0.50–1.17)	
Coronary heart disease						
No	118 (61.1)	136 (48.2)	7.68	0.006	1	
Yes	75 (38.9)	146 (51.8)			1.69 (1.16–2.45)	
Chronic kidney disease						
No	178 (92.2)	249 (88.3)	1.95	0.163	1	
Yes	15 (7.8)	33 (11.7)			1.57 (0.83–2.98)	
Stroke						
No	176 (91.2)	218 (77.3)	15.62	<0.001	1	
Yes	17 (8.8)	64 (22.7)			3.04 (1.72–5.38)	
Anxiety						
No	161 (83.4)	136 (48.2)	60.57	<0.001	1	
Yes	32 (16.6)	146 (51.8)			5.40 (3.46–8.43)	
Notes.

OR odds ratio

95% CI 95% confidence interval

T2DM type 2 diabetes mellitus

Multivariable analysis of factors associated with sleep quality

All variables with a P value of <0.05 in between-group comparisons had a VIF value of <5, indicating the absence of multicollinearity, and were all included in the multivariable analysis. The results of multivariable analysis are shown in Table 5. Age, regular physical activity, stroke and anxiety were independently associated with sleep quality in patients with concurrent T2DM and hypertension. Participants aged ≥70 (aOR = 2.12, 95% CI [1.29–3.48]), and with stroke (aOR = 2.16, 95% CI [1.15–4.06]) or anxiety (aOR = 4.24, 95% CI [2.58–6.98]) were at higher odds of poor sleep quality. Those who had regular physical activity were at lower odds of poor sleep quality (aOR = 0.53, 95% CI [0.34–0.84]). Specifically, those over the age of 70 years at 2.12-fold higher odds of poor sleep quality; those with a history of stroke at 2.16-fold higher odds of poor sleep quality; those with anxiety at 4.24-fold higher odds of poor sleep quality; and those who had regular physical activity at 47% lower odds of poor sleep quality.

Table 5 Multivariable associations of factors with sleep quality.

Variables	b	SE	Wald-test	aOR (95% CI)	P value	
Age (years)						
50–69				1		
≥70	0.75	0.25	8.86	2.12 (1.29–3.48)	0.003	
Sex						
Male				1		
Female	0.40	0.36	1.26	1.49 (0.74–2.99)	0.262	
Marital status						
Unmarried				1		
Married	−0.52	0.34	2.34	0.60 (0.31–1.16)	0.126	
Educational level						
Elementary school or below				1		
Middle school	−0.09	0.28	0.10	0.91 (0.53–1.59)	0.749	
High school or above	0.07	0.39	0.03	1.07 (0.50–2.31)	0.866	
Per capita monthly household income (yuan)						
≤3,000				1		
>3,000	−0.08	0.30	0.07	0.92 (0.52–1.66)	0.791	
Working status						
No working				1		
Working	0.01	0.38	<0.01	1.01 (0.48–2.14)	0.977	
History of smoking						
No				1		
Yes	0.23	0.35	0.43	1.26 (0.63–2.50)	0.512	
Regular physical activity						
No				1		
Yes	−0.63	0.23	7.39	0.53 (0.34–0.84)	0.007	
Duration of T2DM (years)						
<10				1		
≥10	0.20	0.23	0.81	1.23 (0.79–1.91)	0.369	
Coronary heart disease						
No				1		
Yes	0.11	0.23	0.22	1.11 (0.72–1.73)	0.642	
Stroke						
No				1		
Yes	0.77	0.32	5.70	2.16 (1.15–4.06)	0.017	
Anxiety						
No				1		
Yes	1.45	0.25	32.46	4.24 (2.58–6.98)	<0.001	
Notes.

SE standard error

aOR adjusted odds ratio

95% CI 95% confidence interval

T2DM type 2 diabetes mellitus

The ROC curve was presented in Fig. 1. The area under the ROC curve of the established multivariable model was 0.776 (95% CI [0.735–0.818]), indicating good predictive ability. In addition, the Hosmer-Lemeshow test (P = 0.260) indicated favorable model calibration.

Figure 1 Receiver operating characteristic (ROC) curve of poor sleep quality prediction model.

Subgroup analyses results of the multivariable model

Similar results were observed in subgroup analyses (Table 6). The area under the ROC curve ranged from 0.70 to 0.79, and the Hosmer-Lemeshow test indicated favorable model calibration across all subgroups (P > 0.05), suggesting the robustness of the established multivariable model.

Table 6 Discrimination and calibration of the multivariable model across subgroups.

Subgroups	Discrimination	Calibration by Hosmer-Lemeshow test	
	Area under the ROC curve (95% CI)	P value	χ 2	P value	
Sex					
Male	0.75 (0.68–0.81)	<0.001	12.83	0.076	
Female	0.77 (0.72–0.83)	<0.001	11.60	0.071	
Educational level					
Elementary school or below	0.79 (0.73–0.85)	<0.001	9.57	0.214	
Middle school	0.70 (0.61–0.78)	<0.001	4.87	0.561	
High school or above	0.72 (0.61–0.84)	0.001	1.63	0.898	
Number of T2DM complications					
0	0.75 (0.66–0.84)	<0.001	2.84	0.900	
≥1	0.77 (0.72–0.82)	<0.001	12.21	0.057	
Notes.

ROC receiver operating characteristic

95% CI 95% confidence interval

T2DM type 2 diabetes mellitus

Discussion

This study investigated the prevalence of poor sleep quality and its associated factors in patients with concurrent T2DM and hypertension in Hunan, China. Though factors associated with poor sleep quality in patients with T2DM or hypertension alone have been well studied, those with concurrent T2DM and hypertension received less attention. To the best of our knowledge, this is the first study to identify the prevalence of poor sleep quality and its associated factors in individuals with both T2DM and hypertension exclusively. This study found that the prevalence of poor sleep quality in patients with concurrent T2DM and hypertension was 59.4% (95% CI [54.9%–63.9%]), which was comparable with a previous study in patients with concurrent T2DM and metabolic syndrome (59.10%) (Li et al., 2022). However, it was higher than previous studies in patients with T2DM or hypertension alone (Birhanu, Hassen Salih & Abate, 2020; Ayanaw et al., 2022; Li et al., 2020), which could be explained by the synergistic effects of diabetes and hypertension on sleep quality (Surani et al., 2015; Wang, Wang & Zhang, 2024; Almalki et al., 2022; Adzrago, Williams & Williams, 2024). Therefore, given the high prevalence of poor sleep quality found in this study and the negative associations of poor sleep quality with subsequent health outcomes, early identification of those at risk of poor sleep quality is crucial (Miller & Howarth, 2023; Ganidagli & Ozturk, 2023; Chasens et al., 2014).

This study added significantly to the existing body of knowledge by comprehensively assessing the role of socio-demographic and lifestyle characteristics, disease-related factors, and anxiety in poor sleep quality among patients with concurrent T2DM and hypertension. Advanced age was found to be associated with higher odds of poor sleep quality in patients with concurrent T2DM and hypertension, which was consistent with previous studies in different populations (Ayanaw et al., 2022; Li et al., 2022; Mikołajczyk-Solińska et al., 2020). For example, a study of hypertension patients in an Ethiopian hospital found that individuals aged ≥65 were at increased odds of poor sleep quality (aOR = 4.07, 95% CI [2.07–7.97]) (Ayanaw et al., 2022). In addition, a study in T2DM patients found advanced age to be a key risk factor for poor sleep quality (OR = 1.11, 95% CI [1.07–1.15]), with those over the age of 72 years at 11-fold higher odds of having sleep problems (Mikołajczyk-Solińska et al., 2020). Furthermore, data from a Spanish population of 2,144 twins showed a direct and significant correlation between older age and poor sleep quality (OR = 1.05, 95% CI [1.03–1.06]) (Madrid-Valero et al., 2017). This can be explained by the fact that aging is an irreversible physiological process which plays an important role in sleep regulation. On the one hand, the number and function of microglia decrease with aging, and the depletion of microglia disrupts circadian rhythms in brain tissue (Choudhury et al., 2021; Hefendehl et al., 2014; Deurveilher et al., 2021). On the other hand, advanced age is often accompanied by vascular stiffening, and hyperglycemia and hypertension further damage cerebral blood vessels and autonomic nerves, which can weaken sleep homeostasis and exacerbate sleep quality in older adults (Surani et al., 2015; Ma et al., 2024; Seravalle, Mancia & Grassi, 2018). Therefore, clinicians should pay special attention to the sleep quality of patients with concurrent T2DM and hypertension who are aged ≥65, and targeted intervention should be implemented timely.

A previous study by Tseng et al. (2020) showed that moderate-intensity exercise training positively affected sleep quality and cardiac autonomic function. In addition, a study in T2DM patients found that those who took exercise to the point of sweating had lower odds of poor sleep quality (OR = 0.48, 95% CI [0.24–0.94]) (Kuo et al., 2021). Consistently, this study found that regular physical activity can protect against poor sleep quality. Exercise may enhance sleep quality by increasing energy expenditure, promoting endorphin secretion, raising body temperature, and reducing levels of pro-inflammatory cytokines (Hasan et al., 2022; Kapsimalis et al., 2008; Driver & Taylor, 2000; Santos, Tufik & De Mello, 2007). In addition, exercise can not only improve insulin sensitivity and reduce the risk of hypoglycemia at night, but also lower systolic and diastolic blood pressure (Bangsbo, 2024; Chomiuk et al., 2024). A previous network meta-analysis showed that Tai chi, Baduanjin, resistance training, resistance training combined with walking, and muscular endurance training combined with walking significantly improved sleep quality more than usual care, and muscular endurance training combined with walking was identified as the most effective exercise regimen for improving sleep quality in older adults (Hasan et al., 2022). Therefore, physical activities can be considered as promising intervention targets to improve sleep quality, and it is strongly recommended to promote regular physical activities in patients with concurrent T2DM and hypertension in clinical practice.

Consistent with previous studies (Sonmez & Karasel, 2019; Falck et al., 2019), this study found that patients with a history of stroke were at higher odds of poor sleep quality. This could be explained by the fact that the coexistence of diabetes and hypertension can aggravate vascular lesions, which can lead to increased risk of stroke (Ohishi, 2018; Jia & Sowers, 2021). Moreover, sleep architecture is regulated by a complex interaction of multiple mechanisms located in the brainstem, hypothalamus, preoptic area and thalamus, while stroke can impair the central nervous system, leading to changes in brain activity, brain function and sleep structure (Falck et al., 2019; Ferre et al., 2013). For example, a study by Kishimoto et al. (2016) found that individuals with a history of stroke had higher odds of sleep disturbances (aOR = 1.63, 95% CI [1.24–2.15]). Given the negative effects of stroke on sleep quality, clinicians should pay more attention to the sleep quality of those who have a history of stroke, and it could also be deduced that the sleep quality of patients with concurrent T2DM and hypertension may be improved through strengthening the preventive measures against stroke.

The relationships between mental health problems and poor sleep quality have been well-established (Bayrak & Çadirci, 2022; El Kherchi et al., 2023; Mousavi et al., 2023). For example, a study of diabetes patients found that anxiety was associated with poor sleep quality (Bayrak & Çadirci, 2022). Furthermore, a study conducted in Ethiopian patients with hypertension showed that individuals with anxiety were more likely to suffer poor sleep quality (aOR = 1.89, 95% CI [1.16–3.03]) (Ayanaw et al., 2022). Similarly, this study found that anxiety can increase the odds of poor sleep quality in patients with concurrent T2DM and hypertension. The relationship between anxiety and poor sleep quality may be bidirectional. On the one hand, patients with anxiety tended to experience longer sleep latency, more frequent awakenings, prolonged night wakings, and fewer transitions into non-rapid eye movement sleep (Krystal, 2012; First, 2013). On the other hand, long-term poor sleep quality itself is an important risk factor for anxiety symptoms (Okun et al., 2018; Zhao et al., 2025). The coexistence of T2DM and hypertension can lead to further exacerbation of this bidirectional cycle due to disease stressors as well as disturbed physiological mechanisms that adversely affect both mental health and sleep (Zhao et al., 2025; Vilela et al., 2024). Therefore, it is recommended to incorporate anxiety screening into the regular follow-up examinations of patients with concurrent T2DM and hypertension, and the sleep quality may be improved through collaborations with endocrinologists and psychologists.

Although not directly assessed in the present study, blood glucose and blood pressure control were key modifiable factors affecting sleep quality in patients with concurrent T2DM and hypertension. Nocturia, nocturnal hypoglycemia, oxidative stress and autonomic neuropathy caused by high blood sugar may disrupt sleep (Surani et al., 2015; Birhanu, Hassen Salih & Abate, 2020; Knutson et al., 2006). Uncontrolled hypertension can affect endothelial dysfunction and excessive sympathetic nerve activity, thus damaging sleep structure (Seravalle, Mancia & Grassi, 2018; Carnethon & Johnson, 2019). Poor sleep quality can in turn lead to poor control of blood sugar and blood pressure (Cho, 2019; Shibabaw, Dejenie & Tesfa, 2023). Previous studies have indicated that enhanced blood glucose and blood pressure control, and standardized antihypertensive treatment such as angiotensin-converting enzyme inhibitors (ACEIs)/angiotensin II receptor blockers (ARBs), can significantly improve patients’ sleep quality (Cho, 2019). In addition, previous studies indicated that in addition to healthcare access, urban or rural settings can have an impact on sleep quality (Wang et al., 2024; Billings, Hale & Johnson, 2020; Kuriyama, 2024). Therefore, more multi-center studies with comprehensive assessments of factors including blood glucose and blood pressure control measures are warranted.

This study had several limitations. Firstly, this was a cross-sectional study, which limited the ability to determine the causal relationships between the associated factors and sleep quality. Secondly, recall bias may exist when collecting information on smoking and drinking history. Thirdly, the PSQI scale used in this study is a subjective measurement. Compared to objective sleep assessment methods such as actigraphy and polysomnography, the PSQI may introduce misclassification. Finally, this study was conducted in a single-center hospital setting, which may limit the generalizability of the findings. Therefore, future multi-center longitudinal studies should be conducted by using the objective sleep assessment methods to elucidate the directionality of the associations.

Conclusions

Poor sleep quality was highly prevalent in patients with concurrent T2DM and hypertension, with nearly three-fifths suffering from poor sleep quality. Age, regular physical activity, a history of stroke and anxiety status were associated with sleep quality in patients with concurrent T2DM and hypertension. More attention should be paid to those with advanced age and a history of stroke, and the sleep quality of patients with concurrent T2DM and hypertension may be improved through promoting regular physical activities and facilitating collaborations between endocrinologists and psychologists in clinical practice. Future multi-center longitudinal studies should use objective sleep assessment methods such as polysomnography to enhance the clinical use.

Supplemental Information

Supplemental Information 1 Raw data

Supplemental Information 2 STROBE checklist

The authors would like to thank all the participants for their participation.

Additional Information and Declarations

Competing Interests

Author Contributions

Human Ethics

Data Availability

The authors declare there are no competing interests.

Ziling Feng conceived and designed the experiments, performed the experiments, analyzed the data, prepared figures and/or tables, authored or reviewed drafts of the article, and approved the final draft.

Hongying Liu conceived and designed the experiments, performed the experiments, analyzed the data, prepared figures and/or tables, authored or reviewed drafts of the article, and approved the final draft.

Ni Xiong performed the experiments, analyzed the data, prepared figures and/or tables, and approved the final draft.

Leyao Tang performed the experiments, analyzed the data, prepared figures and/or tables, and approved the final draft.

Wenjie Dai conceived and designed the experiments, authored or reviewed drafts of the article, and approved the final draft.

Fang Yang conceived and designed the experiments, authored or reviewed drafts of the article, and approved the final draft.

The following information was supplied relating to ethical approvals (i.e., approving body and any reference numbers):

The study protocol was approved by the Ethics Committee of Xiangya School of Public Health, Central South University (No. XYGW-2021-27).

The following information was supplied regarding data availability:

The raw measurements are available in the Supplementary File.

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
