# Peer review of "Prevalence of poor sleep quality and its associated factors in patients with concurrent type 2 diabetes mellitus and hypertension"

_PeerJ, doi:10.7717/peerj.20325_

## Round 0.1 · original submission · Major Revisions

· Academic Editor

Major Revisions

**Language Note:** PeerJ staff have identified that the English language needs to be improved. When you prepare your next revision, please either (i) have a colleague who is proficient in English and familiar with the subject matter review your manuscript, or (ii) contact a professional editing service to review your manuscript. PeerJ can provide language editing services - you can contact us at [email protected] for pricing (be sure to provide your manuscript number and title). – PeerJ Staff

Reviewer 1 ·

Basic reporting

Title & Abstract
Title:
The title specifies the patient group being evaluated; it also highlights the primary emphasis of this research. Nonetheless, it could have been made more descriptive with minimal rephrasing. The expression “type 2 diabetes mellitus patients with hypertension” is somewhat redundant and does not wholly justify the significance of both chronic diseases in the study. A more precise phrasing would be “Prevalence of Poor Sleep Quality and Its Associated Factors in Patients with Concurrent Type 2 Diabetes Mellitus and Hypertension.” This version preserves brevity while highlighting poor sleep quality, and its associations as plausible comorbid characteristics of T2DM and hypertension.
Abstract
The abstract succinctly summarizes the study's aims, methodologies, and principal findings. The incorporation of metrics like prevalence statistics, and an Area Under the Curve value enhances its informational significance. The abstract methodology section should explicitly state which evaluation tools were employed (such as the Pittsburgh Sleep Quality Index and GAD-7), readers’ easy access to this information improves the study’s replicability. Furthermore, the chosen sampling or recruitment strategy can be succinctly indicated in the abstract (under methodologies).

Introduction
The introduction provides an overview of the epidemiologic data on diabetes and hypertension, both regionally and worldwide. It also offers perspective on the growing acknowledgment of sleep quality as a determinant factor in chronic disease outcomes. The authors offer justification for investigating synergistic effects of concurrent T2DM and hypertension on sleep quality. Other research substantiating the pathophysiological and clinical significance of these diseases provide added context to the relevance of the study.
The introduction can be enriched by including a more extensive analysis of the processes through which a causal relationship might exist between poor sleep quality, T2DM and hypertension; especially emphasizing bidirectional relationships involving inflammation, autonomic dysfunction, and metabolic dysregulation. The authors briefly refer to these factors in the second paragraph of the "Introduction," however, further description of the aforementioned factors would enhance readers’ understanding of their relationship. Moreover, the introduction does not articulate whether a distinct knowledge gap exists from prior research, e.g. overlooking patients with concomitant T2DM and hypertension. If so, a statement verifying the rarity of dual-condition studies should be appended to the conclusion of the third paragraph in order to underscore the validity of this research.
To enrich the literature review, I suggest the authors include the following citation on page 2, at the end of the third paragraph below the “Introduction” heading:
Donga et al., 2010 https://pubmed.ncbi.nlm.nih.gov/20357381/ : an overview of this publication shows the metabolic implications of sleep deprivation with respect to enhancing the correlation between sleep quality and glucose metabolism. Including this reference in the work will add vital comprehension of the subject matter as it corresponds with the study's aims.
Figures & Tables
The tables and figures are properly formatted, displayed and sufficiently labeled. The Receiver Operating Characteristics curve is useful as it improves the understanding of the logistic regression model. Tables 1 to 5 are systematically arranged, providing data summaries and details regarding the variables collated. All tables must however contain footnotes that clarify any abbreviations used.

Experimental design

Material and Methods
The Methods section is clear, and it is determined that a cross sectional study design would be used. Both the inclusion and exclusion criteria are aptly stated within the "Study Design and Population" paragraph. The period from July 2022 to February 2023 is plausible for obtaining a substantial sample size of 475 patients. Although the sample size is sufficient for conducting statistical analysis; the absence of sample size calculation to ascertain statistical power undermines its scientific rigor.
The research instruments used to assess anxiety and sleep quality (GAD-7 and PSQI) have been well-validated in previous studies, and the authors have rightly gone further to acknowledge their validation in an Asian population. The operational definitions of smoking, alcohol consumption, and physical activity are clear enough to facilitate replicability. In my opinion that the authors should incorporate a more comprehensive elucidation of the method used in administering questionnaires (e.g., interviewer-administered versus self-administered), while also indicating whether illiterate or physically-impaired individuals were assisted with filling the questionnaires. This information should be incorporated into the second paragraph under “Data Collection.”
The statistical test carried out are reasonable considering the data collected, and the derivation of a logistic regression statistic is fitting for validating separate connections. Nevertheless, this paper lacks an explanation regarding the management of absent data. A statement must be inserted beneath the "Statistical Analysis" header, before the concluding sentence, to highlight the percentage of missing data and strategies employed for their imputation or exclusion. The authors should specify the software and any packages utilized for calculating the Receiver Operating Characteristics curve.

Validity of the findings

Results
The results section is well written and its outcomes are articulated clearly and align with the methodology. The prevalence of inadequate sleep quality (59.4%) is significant and accompanied with a confidence interval, which enhances the result’s reliability. The findings of multivariable logistic regression are provided clearly, accompanied by relevant odds ratios and confidence interval ranges. The authors have effectively addressed the ROC curve (AUC = 0.771), indicating that this model ultimately had a good performance.
The contribution from results is small, yet significant. This study is among a few which investigated Chinese populations with comorbid T2DM and hypertension; despite the existence of similar studies that assessed either of the aforementioned diseases independently. The study’s novelty resides in population and regional specificity, rather than in statistical methodologies or metrics employed. Nevertheless, the data is credible and aligns with findings in analogous groups.
A possible constraint remains the absence of comprehensive information regarding study participant recruitment strategy. Determining which ideal sampling method was employed would affect the generalizability of the results. This should be added to the initial paragraph in the section titled “Study Design and Population.” Moreover, considering that more than 74% of the patients experienced at least one complication of T2DM, could a subgroup analyses have given valuable insights into whether these secondary issues influenced the relationship with sleep quality.

Discussion
The ‘Discussion section’ is systematically structured and corresponds with results of the investigation. The authors accurately interpret the data and relate their findings to prior similar investigations. The discourse on anxiety, stroke, physical activity, and age as correlating factors is substantiated by pertinent citations. Nevertheless, the authors may provide additional detail on how these correlations may guide clinical screening or therapies.
A notable deficiency lies in the absence of discourse regarding the potential influence of regulating hyperglycemia and hypertension, both of which are pertinent and clinically modifiable factors. Incorporating this into the third paragraph under the "Discussion" section would enhance the practical significance of the findings. The limitations as state dare logical; however, the inability to use objective sleep assessments such as actigraphy or polysomnography suggests that more discussion Acknowledging the potential influence of subjective bias on prevalence estimates.
For enhanced contextualization, the subsequent reference could be cited on page 8, at the conclusion of the fourth paragraph under the "Discussion" heading: Grandner et al., 2016 to elucidate the impact of sleep disorders on cardiovascular and metabolic disease risk. Adding this citation may serve to enhance the biological plausibility of the author’s findings and expand their discourse.

Conclusion
The conclusions align with the study results and concisely reiterate the principal findings. The authors purport implications for clinical care, and advocate for the implementation of physical activity and anxiety screening as realistic and effective interventions among patients with similar disease characteristics as the study participants. Nonetheless, the conclusion may be enhanced by including a statement that underscores the necessity for longitudinal research to elucidate the directionality of these correlations. This additional statement should be placed subsequent to the final phrase under the "Conclusions" heading.
Moreover, although the study’s findings warrant advocating for targeted screening and possible intervention measures, the authors’ recommendations might be enhanced by proposing collaboration between endocrinology and mental health services to comprehensively address sleep quality. Incorporating such integrated suggestion would augment the clinical significance of the findings. This study examines a significant and inadequately investigated subject in research on chronic disease comorbidity. The significant prevalence of suboptimal sleep quality among patients with concurrent T2DM and hypertension poses a public health concern; the study provides essential data from a Chinese clinical cohort. The methodology is robust, and the application of valid tools and instruments enhances the reliability of the study’s findings. The manuscript would improve with a more explicit discussion of absent data, participant recruiting strategy, and possible effects of appropriate blood pressure and glycemic control. Furthermore, an enhanced integration of mechanistic insights and public health consequences would amplify the significance of this study’s findings. This manuscript, with minimal changes, could significantly contribute to existing knowledge on chronic disease associations with sleep quality. The manuscript needs to be edited by a native English doctor to enhance the grammar and flow of the text.

·

Basic reporting

The manuscript follows a clear structure and provides adequate background with relevant literature. Figures and tables are appropriate and well-labeled, and raw data is shared. The results align with the stated aims. However, the writing would benefit from minor language editing for clarity. Additionally, the authors' names and institutional affiliations are visible, which violates the anonymity standard for peer review and should be corrected.

Experimental design

The study presents original research that fits the journal’s scope and addresses a relevant gap by focusing specifically on T2DM patients with hypertension. The research question is clear and meaningful. Ethical approval is obtained, and inclusion/exclusion criteria are well defined. Data collection methods and tools are appropriate and replicable. However, the single-site design may limit generalizability and should be more explicitly acknowledged.

Validity of the findings

The data are complete, statistically sound, and appropriately analyzed. The conclusions are clearly stated and aligned with the research question and results. The ROC analysis supports model validity.

·

Basic reporting

The manuscript discusses an important clinical issue: the prevalence and correlates of poor sleep quality in patients with both type 2 diabetes mellitus (T2DM) and hypertension. The dataset is large (n = 475), and the statistical analyses are generally appropriate. The study offers novel insights by focusing specifically on patients with dual diagnoses. However, the manuscript needs significant revisions in several areas before it can be considered for publication.

Experimental design

The study uses appropriate methods, but the following improvements are needed:
- Provide a power/sample size calculation.
- Include model calibration statistics.
Consider testing interaction terms, such as anxiety × exercise.

Validity of the findings

Results are generally well presented. Consider adding:
- Effect sizes and CIs in univariable results.
- Optional: stratified analyses by sex or education level.

The discussion, although well-referenced, would benefit from a stronger connection between the findings and clinical practice.

Additional comments

The manuscript should undergo professional English editing. Common issues include grammatical errors and awkward phrasing. Examples:
- Line 18: "The coexist of..." → "The coexistence of..."
- Line 25: "to identity..." → "to identify..."
- Line 33: Remove the article "a" before "lower odds."

Expand the limitations section:
- Lack of objective sleep metrics. The exclusive use of subjective measures (PSQI and GAD-7), without objective sleep data or medication history, should be a limitation in interpretability.
- No data on medications.
- Limitations of generalizability from a single hospital sample.

---

## Round 0.2 · Minor Revisions

· Academic Editor

Minor Revisions

**Language Note:** When preparing your next revision, please ensure that your manuscript is reviewed either by a colleague who is proficient in English and familiar with the subject matter, or by a professional editing service. PeerJ offers language editing services; if you are interested, you may contact us at [email protected] for pricing details. Kindly include your manuscript number and title in your inquiry. – PeerJ Staff

Reviewer 1 ·

Basic reporting

Title & Abstract
The title is descriptive and accurately represents the study's substance and emphasis. The primary variable (poor sleep quality), methodological objective (evaluation of associated factors), and patient group (with hypertension and diabetes) are carefully delineated. The abstract offers a thorough overview of the context, results, methods, and conclusions.

Introduction
The introduction has been improved. The authors report the significance of sleep quality in this framework, the epidemiological importance of diabetes and hypertension, and the information gap in these patients. The added references regarding the pathophysiological causes of inadequate sleep are useful.

Figures & Tables
The tables are adequate, systematically presented, and suitable for displaying results. Table footnotes adequately show the information about statistical significance and abbreviations. The ROC curve is pertinent and has correct labeling and data interpretation.

Experimental design

Material and Methods
The techniques section has been enhanced and provides adequate details. The inclusion and exclusion criteria, sampling strategies, and the instruments employed are validated and suitable for this cohort. The time period is sufficient for gathering pertinent data. The statistical methods have been accurately implemented and carefully chosen.

Validity of the findings

Results
The modified manuscripts present results that are distinctly reported. Tables are suitably and the results are rational and coherent. The multivariable regression model is meticulously organized.

Discussion
This section nicely synthesizes the results with pertinent literature. The study nicely compares the findings with previous research, including data from different geographic situations. Mention of physical activity as a modifiable factor is expected, but scientifically shown to be a significant advantage. The paragraph about blood pressure regulation and hyperglycemia is without direct evidence from the study, which is recognized as a limitation. This paragraph could start with a sentence, like "Although not directly assessed in the present study..." to ensure proper scientific interpretive rigor. The authors can mention whether the rural or urban settings, in addition to healthcare access, can have an impact on sleep quality, even if these factors were not assessed.

Conclusion
The conclusions adequately correspond with the findings in addition to providing pertinent clinical implications. The focus on modifiable risk factors like anxiety and physical exercise underpins the development of future management strategies. The need for multi-center longitudinal studies is mentioned; however, the authors can improve this by mentioning that objective sleep measures, such as polysomnography or actigraphy, could enhance the clinical use of future research.

Additional comments

The authors have sufficiently addressed the issues raised in the review.

The manuscript still has wording inconsistencies and minor grammatical errors. The manuscript needs to be edited by a native English speaker to enhance the grammar and flow of the text.

·

Basic reporting

This is a well-structured and clearly written manuscript addressing an important public health concern.

Experimental design

-

Validity of the findings

-

·

Basic reporting

The authors have addressed most of the recommendations, and the manuscript has improved. However, several areas need improvement before the manuscript can be considered for publication in PeerJ.
While the topic is relevant, the novelty is moderate because similar work exists in other populations. The discussion should emphasize how this study advances knowledge beyond prior publications. A stronger contextualization of the findings within recent literature from 2022–2024 is required.

The authors should report exact p-values instead of thresholds. Interpret the clinical significance of aOR values. Add 95% CIs to AUC in Figure 1. Ensure consistent formatting in tables. Please review the PeerJ guide to authors. Include mean ± SD for PSQI alongside poor sleeper prevalence and present gender distribution earlier.

The cross-sectional design is correctly stated, but limitations for inferring causality should be explicitly highlighted in both the abstract and discussion. Clarify operational definitions: physical activity, anxiety cutoff on GAD-7, and data source for comorbidity duration.

Experimental design

-

Validity of the findings

-

Additional comments

In the discussion section, link mechanisms more directly to T2DM/hypertension pathophysiology. Avoid speculation. Mention the bidirectional relationship between poor sleep quality and anxiety.

---

## Round 0.3 · accepted · Accept

· Academic Editor

Accept

One reviewer still has several comments on your manuscript. Please revise it accordingly. Also, our staff mentions several editorial and language suggestions. Please check it and make it perfect before publication.

·

Basic reporting

The authors have satisfactorily addressed the majority of my previous recommendations, particularly by expanding the methodological details, reporting statistical analyses more comprehensively, and strengthening both the discussion and the limitations section. The manuscript is now substantially improved. At this stage, the only remaining issue is to carefully review the grammatical structure and phrasing of specific sentences to ensure clarity and fluency in English throughout the text. See lines 227-233.

Experimental design

-

Validity of the findings

-